# Regulation of Structure and Quality of Dried Noodles by Liquid Pre-Fermentation

**DOI:** 10.3390/foods10102408

**Published:** 2021-10-11

**Authors:** Xiaoqing Xiong, Chong Liu, Xueling Zheng

**Affiliations:** College of Grain and Food, Henan University of Technology, Zhengzhou 450001, China; xiaoqingxiong@163.com (X.X.); xuelingzheng@haut.edu.cn (X.Z.)

**Keywords:** liquid pre-fermentation, dried fermented noodles, yeasts, microstructure, quality, flavor

## Abstract

Liquid pre-fermentation technology was innovatively applied to the development of dried fermented noodles. The effects of fermentation time (1, 3 and 6 h) and yeast addition (0.2, 0.5 and 1.0 g/100 g of flour) on the quality, microstructure and flavor of dried noodles were also investigated in this study. Conspicuous porous structures and greater thickness of dried noodles were found when the fermentation time was ≤ 3 h and the yeast addition was ≥ 0.5 g/100 g of flour, which contributed to the increase in the breaking strength, cooking time and water absorption. However, when the fermentation time increased to 6 h, finer microporous structures, little change related to thickness and richer flavor levels were detected. Additionally, the total titratable acidity of dried fermented noodles was increased to 3.38–4.43 mL compared with the unfermented noodles (2.15 mL). Weaker gluten network structures caused by long-time fermentation and acidic environment led to lower hardness, chewiness, tensile force and tensile distance of cooked fermented noodles.

## 1. Introduction

Dried noodles, as the representative of Chinese traditional noodle products, have become an important sector in the food industry [1]. The popularity of dried noodles, particularly in Asian countries is increasing for their simple preparation, desirable sensory attributes, long shelf life augmented with product diversity and nutritive value [2]. As the world market is expanding, the demands of consumers for some distinctive dried noodles products are increasing, especially for fermented dried noodles [3]. However, few dried fermented noodles were found in the market.

As a fermented noodle product, hollow noodles have been popular in China for hundreds of years [4]. At present, hollow noodles are mainly handmade and spontaneously fermented based on previous experience by a complex microbiome dominated by the yeast population. Therefore, the manufacturing procedures of handmade hollow noodles are various, complicated and time-consuming [5]. New fermented dried noodle products and innovative methods for preparing dried fermented noodles should be further studied.

Direct fermentation and indirect fermentation are often used in the production of bakery products. Direct fermentation is to mix all the starter, flour, water and additives and then the mixture is fermented at 27–30 °C for 1–3 h [6]. Indirect fermentation is to mix a portion of flour, a portion of water and starter into a dough and allow it to ferment for a specific time; this is the “sponge stage” [6,7]. After fermentation, the sponge and the rest of the ingredients are mixed. The texture and aroma of bread with direct fermentation are inferior to those of indirect fermentation because of the shorter fermentation time. However, indirect fermentation technology can regulate or prolong the fermentation time, improve the flavor and control the qualities of product easily [8].

The indirect fermentation technology includes the traditional sourdough and pre-fermentation technology. The use of sourdough technology as a form of leavening is one of the oldest biotechnological processes in bread production [9]. However, in order to improve the nutritional, functional and sensory characteristics of noodle products, sourdough fermentation as an innovative strategy was recently applied to the process of pasta-making [10,11,12]. It is noteworthy that the long fermentation time of sourdough technology (about 12–24 h) and organic acids produced by *Lactobacillus* contributed to a higher acidity and weaker gluten network structures of the noodles and therefore the sourdough noodles tended to be sour, sticky and soft and had low acceptability [12,13]. Specifically, the increased intramolecular electrostatic repulsion caused by sizeable positive net charges in an acidic environment led to an unfolding of the gluten proteins and an increased exposure of hydrophobic groups [9]. Moreover, the presence of strong intermolecular electrostatic repulsive forces prevented the formation of new bonds [14]. However, the fermentation time of pre-fermentation technology is less than the sourdough fermentation and yeast is used as a starter. The pre-fermentation has widely been employed in bread production. Breads prepared by pre-fermentation are said to exhibit superior flavor, less energy to leaven and higher bread volume than those of straight dough [15,16,17,18]. Ayati et al. [16] reported that the maximum yeast survival was observed at a short pre-fermentation time (30 min), which led to a more desirable bread. Gassenmeier and Schieberle [19] showed that French-type wheat bread prepared by liquid pre-fermentation exhibited a more typical flavor. There is no report on the application of pre-fermentation technology in the production of noodles at present.

Considering pre-fermentation technology can avoid the high acidity of sourdough and give the dough a special flavor, this fermentation method may have the potential to make new-type fermented noodles. Therefore, pre-fermentation technology has innovatively been applied to the manufacturing process of dried fermented noodles.

The aim of this study was to explore the feasibility of liquid pre-fermentation technology applied to noodle-making and investigate the impact of the fermentation time (1, 3 and 6 h) and the amount of yeast added to pre-fermented batter (0.2, 0.5 and 1.0 g/100 g of flour) on the quality, microstructure and flavor of dried noodles. This study might provide a new idea for the development of dried fermented noodle products.

## 2. Materials and Methods

### 2.1. Materials

Wheat flour was obtained from Wudeli Flour Group Co. Ltd. (Handan, China). The moisture content (13.11%), ash content (0.41%) and protein content (10.91%, on a 14.00% moisture basis) of the flour were determined according to the AACC 44–40, 08–01 and 46–08 [20] approved methods. Pressed yeasts were purchased from Angel Yeast Co., Ltd. (Chongzuo, China).

### 2.2. Dried Noodles Preparation

Unfermented noodles were prepared with 400 g of wheat flour and 132 g of distilled water in a Kitchen Aid 5KPM5 planetary mixer (KitchenAid Europa, Inc., Brussels, Belgium) equipped with a flat beater. First, the dough crumbs were mixed at a mixing speed of 60 rpm/min (speed 1) for 1 min, then the mixing speed was changed to 148 rpm/min (speed 5) and mixing for 3 min and finally, the dough crumbs were mixed at a mixing speed of 104 rpm/min (speed 3) for 3 min. After mixing, the prepared dough was placed in a plastic bag and rested for 30 min at 25 °C. The noodle dough was then passed through a laboratory JMTD 168/140 noodle machine (East Fude Technology Development Center, Beijing, China) at the gap of 3 mm for 4 passes. Each pass included folding the dough sheet to double its thickness. This process promoted the wet gluten in the dough to form a fine network structure and distribute it evenly. After being double-folded, the roll gap was adjusted to 2.0 mm for another pass. Following that, five more passes were made to reduce the gap progressively to 1 mm, which made the gluten network structure more compact. Finally, the obtained sheet was cut into rectangles of dimensions of 50 cm × 0.2 cm × 0.1 cm and underwent the noodle drying stage. The resultant noodle strands were hung on rods and placed in an SP-18S incubator (Jiangsu Sanmai Food Machinery Co., Ltd., Jiangsu, China) at 38 °C and 85% RH for 2 h, then the relative humidity was reduced to 65% (10 h) and finally they were dried at room temperature for 4 h.

For the fermented noodles, the preparation of pre-fermented batter was performed as previously described by Moroni et al. [13]. Briefly, varying amounts of dried yeasts were suspended in 96 g of distilled water and incubated at 35 °C for 10 min, then 80 g of flour (20 g/100 g of flour) was mixed with the yeast aqueous solution to give a dough yield (DY) of 220. The added amounts of yeast were 0.2, 0.5 and 1.0 g/100 g of total flour, respectively. After 1, 3 and 6 h of fermentation at 35 °C, the pre-fermented batter, the remaining flour (320 g) and water (36 g) were mixed in a 5KPM5 mixer. This meant that the total amount of water was 33% in all noodle doughs. The subsequent processes of noodle-making and drying were the same as the above steps. After drying, the dried fermented noodles were cut into a length of 20 cm and stored in sealed plastic bags for the next studies. Noodles were prepared on two different days (2 independent trials).

The pre-fermented batters and dough crumbs were sampled in clean beakers for the subsequent test analysis. A portion of dried noodles was ground into powder and sieved by an 80-mesh sieve for later use.

### 2.3. pH and Total Titratable Acidity (TTA)

The pH and TTA of the batters, noodle doughs and dried noodles were determined according to the method of Fois et al. [12]. Both pH and TTA were determined with a pH-meter model Cyberscan 510 (Lennox, Dublin, Ireland), after homogenization of 10 g of sample in 90 mL of distilled water. After 10 min gentle stirring of the batters, noodle doughs and dried noodles, the pH was determined and the samples were titrated to a pH of 8.5 with 0.1 N NaOH. The TTA was reported as mL of NaOH per 10 g of sample. The results were the average value of the two measurements.

### 2.4. Dough Sheet Color

The determination of the color of dough sheets was performed using a Satake mini color grader (MICG1A, Satake Inc., Hiroshima, Japan) according to the method proposed by Fu et al. [21] with some modifications. The colorimeter mode was adjusted to a dry basis state and the illuminant was derived from a pulsed-xenon lamp. The same dough sheet, rectangular with dimensions of 10 cm × 10 cm × 0.1 cm, was measured three times in different positions and the L* (lightness), a* (red-green) and b* (yellow-blue) values were recorded. ∆E was a single value that took into account the color differences between the fermented dough sheets (L*_1_, a*_1_ and b*_1_) and the control sample (L*_0_, a*_0_ and b*_0_):(1)$$\Delta E=\sqrt{{\left(L^{*}{}_{1}-L^{*}{}_{0}\right)}^{2}+{\left(a^{*}{}_{1}-a^{*}{}_{0}\right)}^{2}+{\left(b^{*}{}_{1}-b^{*}{}_{0}\right)}^{2}}$$

The dough sheets were prepared on two different days (2 independent trials). The average value was taken as the final result.

### 2.5. Microstructure

The microstructure of dried fermented noodles was examined by a Quanta 250 FEG scanning electron microscope (FEI, Hillsboro, OR, USA). Dried noodles were cut in cross-sections (about 1.0 cm × 0.3 cm × 0.2 cm) and separately placed on the sample holder with the help of a double-sided Scotch tape and sputter-coated with gold before being transferred to the microscope [22]. The acceleration voltage was 10 kV and the micrographs of scanning electron (SEM) microscope were taken at 220 and 1000 times magnification.

### 2.6. Mechanical Properties

The mechanical properties of dried noodles were determined by referring to the method of Han et al. [23] with slight modification. The A/SFR probe was selected and the breaking strength and break distance of dried noodles were measured by the compression mode of a TA-XT 2i texture analyzer (Scarsdale, NY, USA; Stable Micro Systems, Surrey, UK). Ten dried noodles with the same length (20 cm) were randomly selected from each experimental batch. The breaking strength of the dried noodles was taken as the maximum force of the curve. The distance to breakage (break distance) gave an indication of the brittleness of the dried noodles. The average of 10 replicate values was reported.

### 2.7. Cooking Properties

Cooking properties were determined according to the method of Liu et al. [24] with slight modification. Twenty strips of noodles with 20 cm length were weighed and cooked in 400 mL of boiling water. Optimum cooking time was expressed as the time when the white core of the noodles disappeared during cooking. The dried noodles, cooked to the optimum cooking time, were drained for 5 min and weighed immediately. Correspondingly, the cooking water was dried to constant weight at 105 °C. Cooking loss was calculated as a percentage of the weight of dry solids in the cooking water to the dried noodles weight. Water absorption was expressed as the percent increase in the weight of the cooked noodles to the uncooked noodles weight.

### 2.8. Texture Properties

Noodles were cut into 10 cm in length for a compressive test (20 strips) or 20 cm in length for a tensile test (12 strips) and cooked in 500 mL of boiling water to the optimum cooking time [25]. Compressive (texture profile analysis, TPA) and tensile tests (within 5 min after cooking) were determined by using a TA.XTPlus Texture Analyzer (Stable Microsystems, Godalming, UK) with the HDP/PFS and A/SPR probes. Each sample was measured five times. The maximum and minimum values were removed and the result was the average of the remaining values.

### 2.9. Volatile Compounds in Dried Noodles

The determination method of volatile compounds in the dried fermented noodles was based on the report of Kim et al. [26]. Gas chromatography–mass spectrometry (GC–MS) analysis was performed on an Agilent Technologies 7890A GC system coupled to an Agilent Technologies 5975C Mass Spectrometer. Five grams of noodle powder were heated to 60 °C in a vial and the headspace was sampled with a 100 µm polydimethylsiloxane fiber (Supelco, Bellefonte, PA, USA) for 40 min. Immediately after that, the fiber was inserted in the GC–MS equipped with the DB-Wax column (30 m × 0.25 mm × 0.25 µm, l × id × ed) (Agilent Technologies, Santa Clara, CA, USA). The temperature of the inlet and interface were both 250 °C. The carrier gas was helium (0.8 mL/min) and the injection mode was splitless. The temperature was 40 °C for 2 min, increased to 180 °C at a rate of 5 °C/min and then to 250 °C at a rate of 10 °C/min. The MS was carried out in electron ionization source mode at 70 eV, the ion source temperature was 230 °C, quadrupole temperature was 150 °C and scanned mass (m/z) ranged from 33 to 450.

### 2.10. Statistical Analysis

The bar diagrams were drawn using Origin 8.5. Statistical analysis was performed with SPSS 19.0 (SPSS Inc., Chicago, IL, USA) software. Statistical analysis of the data related to pH, TTA, color, mechanical properties, cooking properties and texture properties of the samples prepared by using different fermentation times and yeast additions were performed by using two-factor (fermentation time × yeast addition) analysis of variance (ANOVA). After that, a one-factor ANOVA for statistical evaluation of the data was used. Differences were considered at a significance level of 95% (*p* < 0.05) by Duncan’s Test.

## 3. Results and Discussion

### 3.1. pH and TTA

pH and TTA values of batters, noodle doughs and dried noodles are shown in Figure 1. The pH values of the pre-fermented batters varied from 5.4 to 5.7, which was much higher than the traditional sourdough (3.5–4.3) [27]. After fermentation for 6 h, the pH of the pre-fermented batter with 1.0% yeast addition (5.50) was significantly higher than that of 0.2% (5.44) and 0.5% (5.42) (see Figure 1A), which might be due to the decrease of fermentable sugar in the batter after a long fermentation time and the inability of the yeast to continue to produce organic acids [28,29,30]. As shown in Figure 1C,D, the pH of noodle dough with pre-fermented batter was significantly lower than the control noodle dough (6.30), while it was opposite for the TTA (1.3 mL). The pH values of noodle doughs decreased with the increase of yeast addition under the same fermentation time. When the yeast addition was invariant, the pH values of batters and noodle doughs fermented for 6 h were lower than those fermented for 3 h and 1 h (see Figure 1A,C). This might be ascribed to the accumulation of organic acids produced by yeasts during fermentation [31]. The two-factor ANOVA results showed that the effects of interaction of fermentation time and yeast addition on the TTA values of all batter and noodle dough samples were significant (*p* < 0.05). In addition, under the same fermentation time, the TTA values of batters and noodle doughs with 1% yeast addition were higher than other additions (Figure 1B,D). As expected, the TTA values of batters and noodle doughs increased with the increase of fermentation time under the same yeast addition, except for the noodle dough with 1.0% yeast. Nonetheless, the 6 h–1.0% noodle dough samples showed the highest TTA (4.75 mL).

The unfermented noodles showed the highest pH (5.76) and the lowest TTA values (2.15 mL). For dried noodle samples, the two-factor ANOVA results showed that the effects of interaction of fermentation time and yeast addition on the pH values were significant (*p* < 0.05). When the fermentation time was invariant, with the increase of yeast addition, the pH values decreased while the TTA values of dried noodles increased (see Figure 1E,F), which was similar to the research results of Kim et al. [26] in steamed breads. When the amount of yeast was 1.0%, the TTA values increased with fermentation time (1 h < 3 h < 6 h). Specifically, the TTA value of the 6 h–1.0% sample reached 4.43 mL.

### 3.2. Dough Sheet Color

Table 1 reports colorimetric data recorded on noodle dough sheets. The two-factor ANOVA results showed that the effects of interaction of fermentation time and yeast addition on the L*, a*, b* and ∆E values of all dough sheet samples were significant (*p* < 0.05). The L* values of all dough sheets added with fermented batters increased significantly compared to controls, which indicated that the lightness of noodles prepared by liquid pre-fermentation method was obviously improved. Besides this, the L* values increased from 88.27 to 94.47, 89.03 to 93.43 and 87.80 to 92.33 with the increase of yeast addition under the fermentation times of 1 h, 3 h and 6 h, respectively. It was found that the L* value of white salted noodles was negatively correlated with protein and wet gluten contents before and after drying [32]. Dowell et al. [33] also indicated that the gluten content was negatively correlated with the L* value of fresh noodles. As discussed above, the more yeasts were added, the lower the pH value and the higher the TTA value of noodle dough (Figure 1C,D). An acid environment (electrostatic repulsion) caused the increase of protein solubility, which led to a loose gluten network structure, enhanced the reflection of light and made the dough sheets brighter [27]. For different yeast additions, the a* values of noodle sheets fermented for 6 h were all significantly higher than other fermentation times. The ∆E value was used to describe the color differences between the fermented dough sheets and the control. As shown in Table 1, the ∆E values of dough sheets with 1.0% and 0.5% yeast added were significantly higher than that of 0.2%.

### 3.3. Microstructure

For microstructure, the SEM images of the dried noodles, both control and samples added with the pre-fermented batters, are shown in Figure 2. Interestingly, compared with the unfermented noodles (Figure 2A,a), dried fermented noodles showed conspicuous porous structures at the cross-section of noodles, when the fermentation time was ≤3 h and the yeast addition was ≥0.5% (Figure 2C,D,F,G). Accordingly, the thickness of dried fermented noodles was increased significantly. However, when the yeast addition was 0.2%, the dried noodles showed microporous structures (Figure 2B,E) and little changes related to thickness were found (Figure 2B,E). During the drying process, yeasts consumed the fermentable sugars and starches of the noodle doughs and produced CO_2_ and ethanol. Additionally, the gluten network of noodle doughs had a certain capacity of gas-holding. The surface of noodles gradually “encrusted” during the drying process, then the gas could not be completely discharged from the noodles, which led to gas cavities with different sizes in the center of the noodles [28,29,34]. The more yeasts were added, the greater the amount of gas was produced and the larger the size of gas cavities was formed. When the fermentation time was longer (6 h), the number and size of pores in the noodles showed little change with the increase of yeast addition (Figure 2H–J). This was mainly due to the logarithmic growth phase of yeast of about 2 h, thus after fermenting for a long time (especially for 6 h), the reproductive ability and gas production of yeast was decreased.

### 3.4. Mechanical Properties

The mechanical properties (breaking strength and flexibility) of dried noodles, as the key parameters for evaluating the dryness of noodle products, are the main texture parameters for noodle quality evaluation [35]. The two-factor ANOVA results showed that the effects of interaction of fermentation time and yeast addition on the breaking strength and flexibility of all dried fermented noodles were significant (*p* < 0.05). As shown in Figure 3A, when the fermentation time of batter was ≤3 h, the breaking strength of dried noodles increased with the increase of yeast addition. When fermented for 1 h and 3 h, the breaking strength of dried noodles with 0.2% yeast was 6.31 g and 6.69 g, respectively, which was close to that of the control (7.25 g). It was also worth noting that the breaking strength reached 13.83 g and 15.30 g with 1.0% yeast and 1 h and 3 h fermentation time, respectively. Furthermore, the flexibility of dried noodles fermented for ≤3 h was significantly lower than that of dried unfermented noodles (Figure 3B). This was probably associated with the conspicuous porous structures and the thickness of dried noodles. When the fermentation time was less than 3 h, large cellular, uncompacted structures and the greater thickness might have caused the dried noodles to be broken with greater force but a lower bending degree (Figure 3A,B).

When the yeast addition was ≥0.5%, the breaking strength of dried noodles fermented for 6 h was significantly lower than that of the dried noodles fermented for ≤3 h, while it was the opposite for the flexibility. The breaking strength of dried noodles with 1.0% yeast was 7.09 g, which was significantly lower than that with 0.2% (9.52 g) and 0.5% (9.09 g) yeast when the fermentation time was 6 h. This phenomenon might be the consequence of significant hydrolysis of the main storage proteins and electrostatic repulsion at the molecular level that prevented the formation of new bonds (a weaker gluten structure) during the noodle-making and drying processes, due to the long fermentation time and acidic environment [36].

### 3.5. Cooking Properties

The results of the cooking properties of the fermented dried noodles are shown in Figure 3C,D. The two-factor ANOVA results showed that the effects of interaction of fermentation time and yeast addition on the cooking time and water absorption of all dried fermented noodles were significant (*p* < 0.05). When the fermentation time of batter was ≤3 h, the cooking time and water absorption of the noodles tended to increase with the increase of yeast addition. This was mainly related to the thickness and the pores of dried noodles. It could be seen from the SEM that the size of porous structures tended to be larger (Figure 2B–G) and the thickness of the dried noodles tended to be greater (Figure 2B–G) with the increase of yeast addition. Porous structures made it easier for water to enter the inside of the noodles and starches could absorb water and gelatinize easily. When the amount of yeast added was less, the microporous structures were formed and the thickness of the noodles did not increase significantly (Figure 2B,E). Therefore, the 1 h–0.2% (227.0 s) and 3 h–0.2% dried noodles (223.5 s) showed less cooking time compared to unfermented noodles (262.50 s). In addition, thicker noodles needed more time to be cooked to their optimal point [37]. Therefore, the greater thickness due to the increase of yeast addition increased the cooking time. For the increase of water absorption, on one hand, the longer the cooking time, the more water the noodles absorbed. On the other hand, starch and protein were not tightly combined for the formation of porous structures, thus water molecules were easily combined with starch and protein, resulting in a great increase in water absorption.

### 3.6. Textural Properties

Textural properties of dried fermented noodles prepared by different fermentation times and yeast additions are given in Figure 4A–D. The two-factor ANOVA results showed that the effects of interaction of fermentation time and yeast addition on the hardness, adhesiveness, chewiness, springiness, tensile force and tensile distance of all cooked fermented noodles were insignificant (*p* > 0.05). The hardness and chewiness of cooked fermented noodles were slightly lower than the unfermented noodles. However, the adhesiveness of fermented noodles was higher than the control noodles (36.63 g.s). This might be related to the degradation of gluten and the hydrolysis of starch caused by fermentation [38]. Zhao et al. [39] reported that the hydrolysis of starches in sourdough mainly occurred in starch amorphous regions with weak binding forces. Under the same yeast addition, the adhesiveness of cooked noodles did not differ significantly with the increase of fermentation time. As shown in Figure 4A,D, there was also no significant difference in the hardness and springiness of fermented noodles (*p* > 0.05). The starches embedded in the gluten network structures of the noodles, which contributed to a good hardness and chewiness of the noodles. When yeast addition was 1.0%, the chewiness of fermented noodles decreased from 2393.86 to 2250.85 g with the increase of fermentation time (Figure 4C), which might be related to the above-mentioned gluten degradation and starch hydrolysis after long fermentation time [36].

Moreover, the tensile forces of fermented noodles were decreased from 24.27 to 22.67 N and 23.50 to 21.07 N with the increase of yeast addition under the fermentation times of 3 h and 6 h, respectively. Regardless of fermentation time, the tensile distance of fermented noodles tended to decrease with the increase of yeast addition (Figure 4F). The tensile distance of fermented noodles prepared by adding batter fermented for 6 h was significantly shorter than that of other fermentation times. In summary, the weakest tensile force and the shortest tensile distance were obtained by the 6 h–1.0% noodles, which might be attributed to the hydrolysis of the gluten network [27].

### 3.7. Volatile Compounds in Dried Fermented Noodles

The detection results of volatile compounds of dried fermented noodles prepared by different fermentation times and yeast addition are shown in Table 2 and Appendix A. A total of 63 volatile flavor compounds were detected in dried fermented noodles. Twelve volatile compounds were identified from the control samples. There were 10, 12 and 17 species detected in dried fermented noodles when the batter was fermented for 1 h and the yeast addition was 0.2%, 0.5% and 1.0%, respectively. When fermented for 3 h, 20, 18 and 20 volatile substances were detected, respectively. When fermented for 6 h, 22, 26 and 24 volatile substances were detected, respectively. It could be seen that the types of volatile substances in dried fermented noodles increased with the increase of fermentation time of the batters.

Esters showed a fruit aroma and were important flavor components for fermented products [40]. As shown in Table 2, the relative contents of esters in dried fermented noodles were significantly higher than that in dried unfermented noodles. The relative contents of esters in 6 h–0.5% and 6 h–1.0% dried noodles were high. Carboxylic acids showed wax, fat and creamy flavor, which could only be detected in dried noodles fermented for 6 h. Olefins had a high aroma threshold, thus they contributed less to the flavor. Nevertheless, the contents of olefins in fermented steamed breads were always the highest, which was consistent with the results of this study [41]. Additionally, the contents of olefins tended to decrease with the increase of fermentation time. 2-pentylfuran, resulting from the Maillard reaction or oxidation reaction of linoleic acid, showed a low aroma threshold and high flavor activity [42]. When fermentation time was ≥3 h, 2-pentylfuran was detected in dried fermented noodles (see Appendix A). Specifically, the contents of 2-pentylfuran were the highest in 6 h–0.2% dried noodles (9.91%). The above results show that fermentation time has a significant effect on the flavor of dried fermented noodles. The richer flavor levels were formed when the fermentation time was longer (≥6 h).

## 4. Conclusions

The present study showed that the liquid pre-fermentation technology can be applied to the production of dried fermented noodles. As long as the fermentation time and yeast addition were properly controlled, the high acidity of noodles could be avoided. When the fermentation time was less than or equal to 3 h, conspicuous porous structures and a greater thickness of dried noodles were found. However, when the fermentation time was increased to 6 h, the amount of yeast added had little effect on the microporous structures of the dried noodles. The liquid pre-fermentation also contributed to form a special flavor for dried fermented noodles and richer flavor levels were obtained when the fermentation time was 6 h. Finally, the effects of liquid pre-fermentation technology on the minerals content, in vitro digestibility of protein and starch in noodles need further study.

## Figures and Tables

**Figure 1 foods-10-02408-f001:**
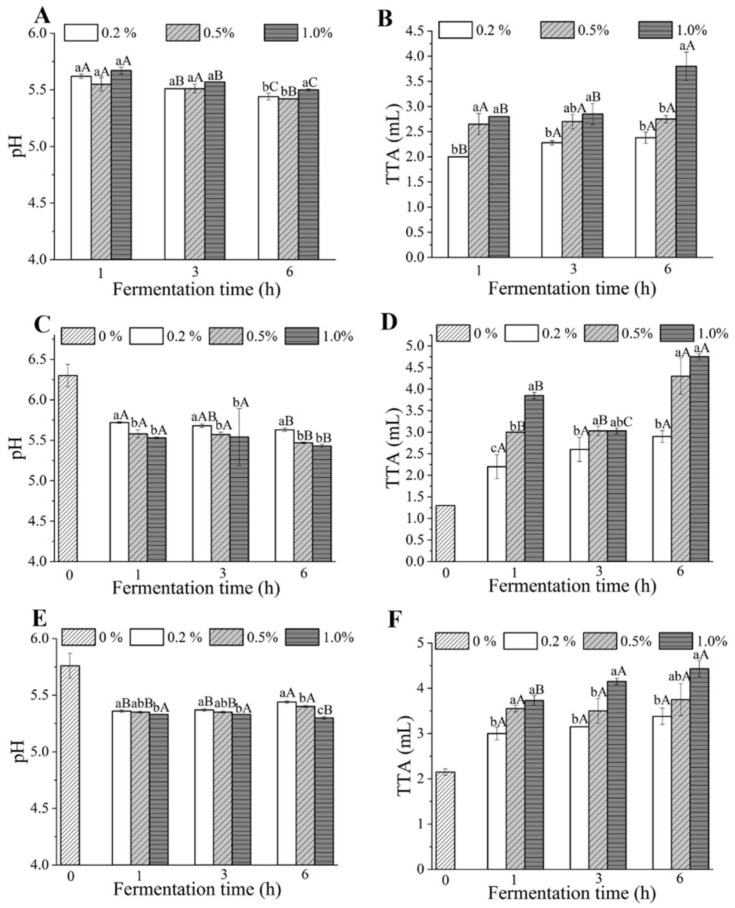
Effect of fermentation time and amount of yeast added on the pH and TTA of batter (**A**,**B**), noodle dough (**C**,**D**) and dried noodle (**E**,**F**) samples. TTA, total titratable acidity. Different lowercase letters indicate that the data related to different yeast additions are significantly different under the same fermentation time. Different capital letters indicate that the data related to different fermentation times are significantly different under the same yeast addition.

**Figure 2 foods-10-02408-f002:**
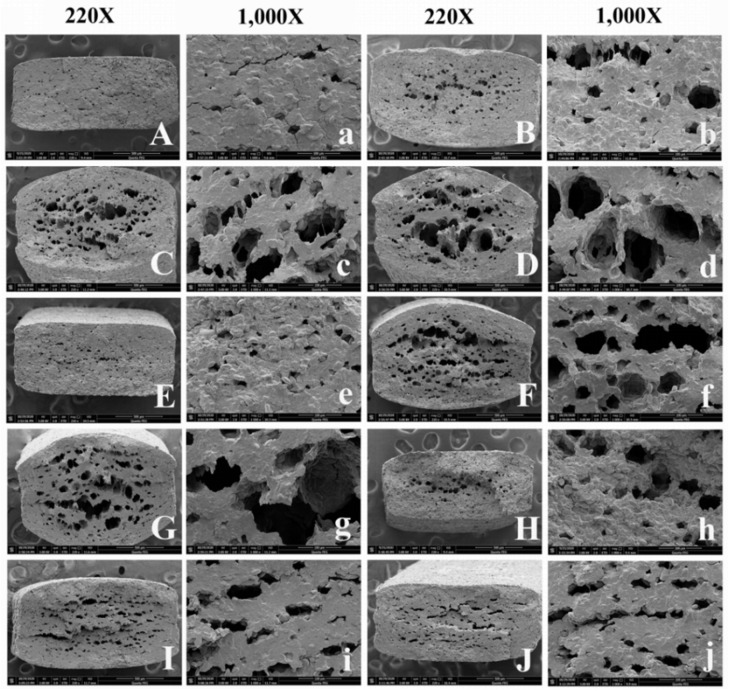
The scanning electron microscope photographs of dried noodles under different fermentation times and yeast additions. Capitalized letters indicate the cross-section of dried noodles magnified by 220 times. Lowercase letters indicate 1000 times magnification. Labels (**A**) and (**a**) represent dried unfermented noodles. (**B**–**D**) and (**b**–**d**) represent a fermentation time of 1 h and yeast addition of 0.2 g, 0.5 g and 1.0 g/100 g of total flour, respectively. (**E**–**G**) and (**e**–**g**) represent a fermentation time of 3 h and yeast addition of 0.2 g, 0.5 g and 1.0 g/100 g of total flour, respectively. (**H**–**J**) and (**h**–**j**) represent a fermentation time of 6 h and yeast addition of 0.2 g, 0.5 g and 1.0 g/100 g of total flour, respectively.

**Figure 3 foods-10-02408-f003:**
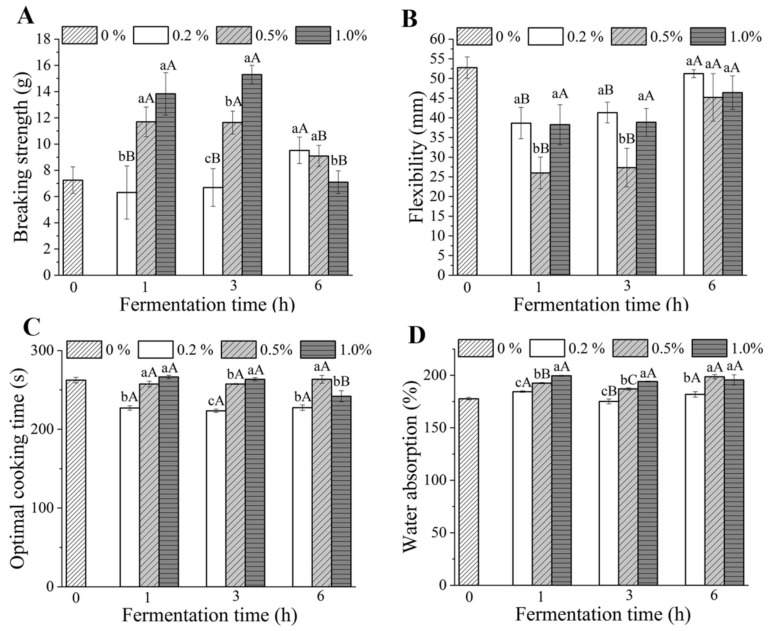
Effect of fermentation time and amount of yeast added on the mechanical (**A**,**B**) and cooking properties (**C**,**D**) of dried noodles. Different lowercase letters indicate that the data related to different yeast additions are significantly different under the same fermentation time. Different capital letters indicate that the data related to different fermentation times are significantly different under the same yeast addition.

**Figure 4 foods-10-02408-f004:**
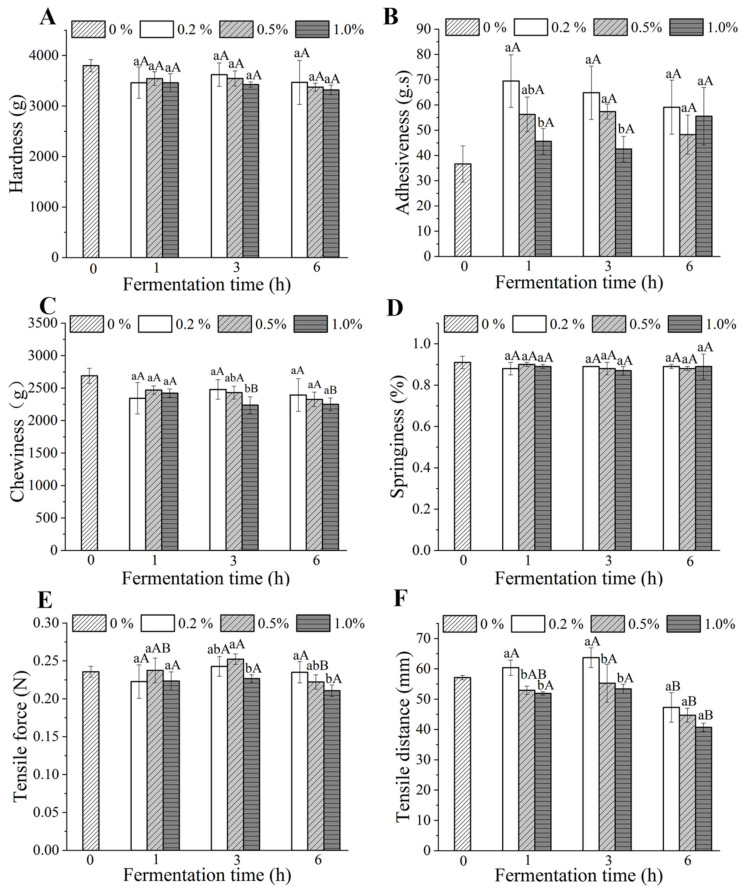
Effect of fermentation time and amount of yeast added on the textural properties of dried noodles. (**A**) Hardness; (**B**) Adhesiveness; (**C**) Chewiness; (**D**) Springiness; (**E**) Tensile force; (**F**) Tensile distance. Different lowercase letters indicate that the data related to different yeast additions are significantly different under the same fermentation time. Different capital letters indicate that the data related to different fermentation times are significantly different under the same yeast addition.

**Table 1 foods-10-02408-t001:** Effect of fermentation time and amount of yeast added on the color of dough sheet.

Fermentation Time (h)	Yeast Addition (%)	L *	a *	b *	∆E
0	0	86.84 ± 0.84	0.27 ± 0.06	21.16 ± 0.30	―
1	0.2	88.27 ± 0.21 ^c,A^	0.30 ± 0.00 ^a,B^	22.00 ± 0.10 ^a,A^	1.66 ± 0.17 ^c,A,B^
	0.5	93.17 ± 0.25 ^b,B^	0.27 ± 0.06 ^a,B^	21.10 ± 0.17 ^b,A^	6.33 ± 0.25 ^b,B^
	1.0	94.47 ± 0.12 ^a,A^	0.20 ± 0.00 ^b,C^	20.77 ± 0.12 ^c,A^	7.64 ± 0.12 ^a,A^
3	0.2	89.03 ± 0.15 ^c,A^	0.20 ± 0.00 ^b,B^	22.13 ± 0.06 ^a,A^	2.40 ± 0.14 ^c,A^
	0.5	93.00 ± 0.10 ^b,B^	0.33 ± 0.06 ^a,A,B^	20.73 ± 0.06 ^c,B^	6.18 ± 0.10 ^b,B^
	1.0	93.43 ± 0.15 ^a,B^	0.27 ± 0.06 ^a,b,B^	21.07 ± 0.12 ^b,A^	6.59 ± 0.15 ^a,B^
6	0.2	87.80 ± 1.37 ^b,A^	0.47 ± 0.12 ^a,A^	21.00 ± 0.36 ^a,B^	1.42 ± 0.68 ^c,B^
	0.5	93.60 ± 0.10 ^a,A^	0.40 ± 0.00 ^a,A^	21.10 ± 0.10 ^a,A^	6.76 ± 0.10 ^a,A^
	1.0	92.33 ± 0.06 ^a,C^	0.40 ± 0.00 ^a,A^	20.93 ± 0.29 ^a,A^	5.50 ± 0.07 ^b,C^

Different lowercase letters indicate that the data related to different yeast additions are significantly different under the same fermentation time. Different capital letters indicate that the data related to different fermentation times are significantly different under the same yeast addition.

**Table 2 foods-10-02408-t002:** Volatile compounds and relative contents of dried noodles under different fermentation times and yeast additions.

		Alcohols	Aldehydes	Esters	Carbonic Acids	Aromatics	Olefins	Furans	Others
0 h–0.0%	Types	1	5	1	0	1	4	0	0
	Relative content (%)	1.07	13.40	0.98	0.00	0.83	73.61	0.00	0.00
1 h–0.2%	Types	0	1	2	0	1	6	0	0
	Relative content (%)	0.00	14.77	9.23	0.00	7.29	68.70	0.00	0.00
1 h–0.5%	Types	0	1	5	0	0	6	0	0
	Relative content (%)	0.00	9.32	11.51	0.00	0.00	78.68	0.00	0.00
1 h–1.0%	Types	0	1	5	0	0	10	0	1
	Relative content (%)	0.00	9.24	26.17	0.00	0.00	58.00	0.00	4.01
3 h–0.2%	Types	2	1	8	0	0	8	1	0
	Relative content (%)	3.10	5.91	16.19	0.00	0.00	63.37	7.66	0.00
3 h–0.5%	Types	0	1	4	0	1	11	1	0
	Relative content (%)	0.00	6.05	12.69	0.00	3.70	66.72	9.24	0.00
3 h–1.0%	Types	0	1	3	0	1	14	1	0
	Relative content (%)	0.00	4.68	6.50	0.00	4.51	70.74	5.30	0.00
6 h–0.2%	Types	0	2	2	1	1	15	1	0
	Relative content (%)	0.00	12.12	4.02	2.73	3.44	54.17	9.91	0.00
6 h–0.5%	Types	0	3	10	1	0	11	1	0
	Relative content (%)	0.00	5.36	47.75	3.83	0.00	39.71	2.28	0.00
6 h–1.0%	Types	1	2	6	1	1	12	1	0
	Relative content (%)	0.50	3.48	55.87	0.42	2.37	29.59	4.68	0.00

Types, the number of volatile substances belonging to a certain class of volatile compounds.

## Data Availability

The dataset of the current study is available from the corresponding authors on reasonable request.

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
