# Peer review of "Regulation of Structure and Quality of Dried Noodles by Liquid Pre-Fermentation"

_foods, 2021, doi:10.3390/foods10102408_

Round 1

Reviewer 1 Report

The authors resubmit a paper on the regulation of structure and quality of dried noodles by liquid pre-fermentation after major revision suggested by the reviewers. The work is interesting and well organized. In particular, the authors have done a careful review of the manuscript and have added details and considerations that can meet even more the needs of the reader. I consider the work ready after minor revisions, suggesting the authors to modify some parts of the text (the section titles could be "Results and Discussion" and "Conclusions") and to do a final check for typos.

Author Response

Thanks to the recommends of Reviewer, we have done a final check for typos:

The “fermented dried noodle” has been revised as “dried fermented noodles ”. Please see line 39-40 in revised manuscript.

The "manufacture" has been revised as "manufacturing". Please see line 77 in revised manuscript.

The "ionisation" has been revised as "ionization". Please see line 197 in revised manuscript.

 The "form" has been revised as "forming". Please see line 452 in revised manuscript.

Moreover, the section titles are "Results and Discussion" and "Conclusions" in our manuscript. However, the format have been changed after uploading: Paragraph 3 is “Results” and Paragraph 4 is "Discussion". We have communicated with the editor on this issue. 

Reviewer 2 Report

After reassessing the manuscript foods-1292513 entitled "Regulation of structure and quality of dried noodles by liquid pre-fermentation" by Xiaoqing Xiong, Chong Liu and Xueling Zheng, I can state that authors made necessary corrections taking under consideration my recommendations.

Author Response

Thanks to the reviewers for their comments and suggestions on the manuscript.

Reviewer 3 Report

This reviewer appreciated the authors' answers to his/her raised criticisms and the new submitted manuscript. However, a mistake in titling paragraphs 3 and 4 was made: paragraph 4 is not a Discussion. Therefore Paragraph 3 has to be titled: Results and Discussion and Paragraph 4 must be titled: Conclusions.

Author Response

The section titles are "Results and Discussion" and "Conclusions" in our manuscript. However, the format have been changed after uploading: Paragraph 3 is “Results” and Paragraph 4 is "Discussion". We have communicated with the editor on this issue.

This manuscript is a resubmission of an earlier submission. The following is a list of the peer review reports and author responses from that submission.

Round 1

Reviewer 1 Report

The authors presents a work on the regulation of structure and quality of dried noodles by liquid pre-fermentation. The paper is interesting and well written/organized, considering all the information given by the authors and the support of the bibliography used. Even with these considerations, the paper in its current form should be made a minor revision of the English language and some adjustments regarding, for example, the methods and conclusion sections are required to facilitate its reading and better clarify its aim and scientific impact in the conclusions that are adequately supported by the data.
Below I propose a list of different comments/corrections to be made in the article, referring to the proof copy.

  1. Introduction Section, page 1, line 23 (note): I suggest to change from "important industry" to " important sector" to avoid repetition in the text;
  2. Introduction Section, page 1, line 34 (note): ".., especially for what concerns ...";
  3. Introduction Section, page 2, line 48 (note): "...which is a difficult method to use...";
  4. Materials and Methods Section, page 2, line 80-81 (note): it may be useful for the reader to include a quote to the AACC International Method;
  5. Materials and Methods Section, page 2, line 91-95 (note): it may be useful to describe the importance of the gap and its subsequent reduction;
  6. Materials and Methods Section, page 3, line 114 (note): the full form of the abbreviation "TTA" is not inserted anywhere in the rest of the text [Total Titratable Acidity];
  7. Materials and Methods Section, page 4, line 160 (note): the use of the p-value is debated [Nature, 567, 305-307 (2019)], so I suggest to insert a brief description of why that specific value was considered and about the lower and upper limit of the intervals considered;
  8. Results and discussion Section, tables (note): meaning of the letters in the table, what is the difference for different values of p? 
  9. Results and discussion Section, page 5, line 188 (typo): remove the bold style;
  10. Conclusions Section (note): I suggest to insert some comments about the microstructure and considerations about the flavor.

Finally, I propose to the authors to better investigate the intermediate time zones between the fermentation hours used in the work in order to have a more complete mapping of the best fermentation conditions. 

Reviewer 2 Report

Review on manuscript: foods-1292513

Regulation of structure and quality of dried noodles by liquid pre-fermentation

by  Xiaoqing Xiong, Chong Liu and Xueling Zheng

submitted to Foods

In the manuscript submitted for comments the authors studied the effect liquid pre-fermentation on the structure and quality of dried noodles.

The manuscript submitted for comments fir well to the aim and scope of the journal, however, it is a pity that the authors did not use sensory methods to evaluate the obtained noodles. The manuscript requires some additions, especially in the methodological part. Also the statistical analysis of the results should be improved.

Detailed recommendations:

line 23 – …industry …industry – the style should be improved,

lines 74-75 – the aim of the study should be clarified, what theoretical support do the authors mean?

lines 78-80 – specific standards numbers should be given,

line 80 – AACC standards are not included in the references,

line 83 – the title of the section suggests different content, and there is also a description of how to prepare unfermented noodles,

line 113 – unexplained abbreviations should not be used in section titles,

line 114 – the abbreviations TTA should be explained during first used,

line 115 – please add a short description of the methods used along with the equipment used,

lines 118-119 – type of illuminant and measurement geometry should be mentioned,

lines 132-138 – what was the sample size, was it conditioned?

line 140 – exactly what? please add a short description of the methods used,

lines 142-142 – what was the sample size, was it conditioned? please add a short description of the methods used,

lines 146-147 – please add a description of sample preparation,

lines 159-160 – with such a planned study design, the statistical evaluation should be based on a two-factor analysis of variance: fermentation time x yeast addition,

Table 1 – the abbreviation TTA should be explained in the legend,

Tables – two-factor anova should be used,

Table 2 – ∆E values in relation to the "0" sample should be calculated and discussed,

Figure 1 – photo descriptions are not always legible,

Table 4 – the meaning of "types" should be explained in the legend,

line 369 – should be: Food & Function,

line 379 – title of journal should be capitalized.

Reviewer 3 Report

In the present study, the authors apply a liquid pre fermentation technology to dried noodles to obtain dried fermented samples. The noodles are investigated at different fermentation times (1-3-6h) and yeast addition (0.2, 0.5 1g/100 g flour) by applying different techniques to monitor pH variation, dough sheet color, microstructure by SEM analysis, mechanical, cooking and texture properties. The results of the study lead the authors to conclude that, with respect to unfermented noodles, the increase of the fermentation time over than 3h together with the increase of the yeast amount affected the mechanical properties of noodles by lowering hardness and chewiness and at the same time increasing adhesiveness after cooking. 

Indeed, the study present a lot of data obtained by different techniques that make the manuscript interesting and complete from a methodologic point of view. However, the data are presented in a very often confusing way and at the same time the English style needs a quite deep revision. Therefore, in this reviewer’s opinion, the manuscript needs a complete reorganization in the following points:

  1. Abstract: No data are reported: the authors only describe the experimental part (not crucial in this section), while do not report resulting data. A vague affirmation can be only found at the end of this section. Moreover, they report that aim of the study is only to improve the flavor of noodles.
  2. This section is generally devoted to analyzing the state of the art as it results from the literature so that the reader can well understand the aim of the study as well as the novelty of the present work. At the end of the section, the aim of the study and a short overview of the results have to be reported. The authors have to reorganize the section in this way, since in this original version, the authors make a mix between literature and personal considerations (see lines 58-63 for example). In particular:

Line 23: Which the meaning?

Line 35 the same

Line 63: investigation, probably

Line 36 and over. English style must be drastically improved.

  1. Results and Discussion. This section must be reorganized both as text and Figures, Tables. The authors present a lot of data. Therefore, they have to be organized to help the reader in easily understanding them. To this aim, first of all, Figures and Tables have to be much more described in the Captions. At the same time, they must be described in detail in the text. For example, Figure 1 reports 20 different and interesting images but in the text it is mentioned only as figure 1. It would be better design the images by capital letters to be able to refer to each of them in the manuscript by the correspondent letter.

Figure 2 is not described in the Caption. At the same time, the labels of the axis are too small.

Table 2. The method has to be much more described otherwise the reported numbers are devoid of meaning. Which the scale, for example?

Table 3. What the authors intend for cooking loss? Again, the caption does not describe the data.

  1. General major remarks: this reviewer is wondering which the real aim of this study. Contradictorily, the same authors report: ‘To improve the flavor’ in the abstract and ‘This study might provide theoretical and technical…Lines 74-75’. In this reviewer’s opinion a study can adopt a lot of methods and then obtain a lot of experimental data, as this one, but if the study does not define a good aim, the data alone have poor meaning. Therefore, the authors have to state well the aim of their study together with the limitations of it. A paragraph devoted to this point is not reported in the manuscript. For example, nothing is reported nor investigated about the nutritional aspect of the study. Is that the content of volatile compounds pays a role in the healthy attribute of this food, just to make an example?
  2. Finally, due to the amount of data at different variables, this reviewer suggests a Figure/scheme where the reader can easily follow the trend of each variable and at the of the flow chart understand the main results of the study.
